# A naturally occurring epiallele associates with leaf senescence and local climate adaptation in *Arabidopsis* accessions

Li He[1,2], Wenwu Wu[1,4], Gaurav Zinta[1], Lan Yang[1], Dong Wang[1], Renyi Liu[1], Huiming Zhang[1], Zhimin Zheng[1], Huan Huang[1], Qingzhu Zhang[1,5] & Jian-Kang Zhu[1,3]

Epigenetic variation has been proposed to facilitate adaptation to changing environments, but evidence that natural epialleles contribute to adaptive evolution has been lacking. Here we identify a retrotransposon, named "NMR19" (naturally occurring DNA methylation variation region 19), whose methylation and genomic location vary among *Arabidopsis thaliana* accessions. We classify NMR19 as NMR19-4 and NMR19-16 based on its location, and uncover NMR19-4 as an epiallele that controls leaf senescence by regulating the expression of *PHEOPHYTIN PHEOPHORBIDE HYDROLASE* (*PPH*). We find that the DNA methylation status of NMR19-4 is stably inherited and independent of genetic variation. In addition, further analysis indicates that DNA methylation of NMR19-4 correlates with local climates, implying that NMR19-4 is an environmentally associated epiallele. In summary, we discover a novel epiallele, and provide mechanistic insights into its origin and potential function in local climate adaptation.

[1] Shanghai Center for Plant Stress Biology and Center of Excellence for Molecular Plant Sciences, Chinese Academy of Sciences, 201602 Shanghai, China. [2] University of Chinese Academy of Sciences, 100049 Beijing, China. [3] Department of Horticulture and Landscape Architecture, Purdue University, West Lafayette, IN 47907, USA. [4] Present address: State Key Laboratory of Subtropical Silviculture, Zhejiang A & F University, 311300 Lin'an, Hangzhou, China. [5] Present address: College of Life Sciences, Northeast Forestry University, 150040 Harbin, China. Li He and Wenwu Wu contributed equally to this work. Correspondence and requests for materials should be addressed to Q.Z. (email: qingzhu.zhang@nefu.edu.cn) or to J.-K.Z. (email: jkzhu@sibs.ac.cn)

Plants are sessile organisms, often challenged by various environmental perturbations. However, they can adapt to their local environments via phenotypic plasticity. Phenotypic diversity induced by genetic variation has a central role in plant adaption[1,2]. Recently, some studies also showed a role for epigenetic variation (e.g., DNA methylation) in short-term as well as long-term evolutionary adaptation[3–10]. The "1001 Epigenomes Project" highlighted a strong correlation between DNA methylation and climate in a global collection of *Arabidopsis thaliana* accessions, further suggesting a potential role for the epigenome in shaping adaptive evolution[11].

Although DNA sequence variation is the primary evolutionary force underlying phenotypic variations, DNA methylation changes may affect the expression of genes and thus can also contribute to trait variations that can be inherited to the next generations[7]. Such stably inherited epigenetic alleles are known as epialleles. Epialleles can lead to variations at the phenotypic and molecular levels, such as flower morphology[12], sex determination[13], fruit ripening[14], starch metabolism[15], vitamin E accumulation[16], plant architecture[17], expression of *PAI* genes[18] and *FOLT* genes[19], flowering time and root length[20,21], biomass[22], and oil palm fruit productivity[23]. In addition, DNA methylation status can be altered by plant exposure to biotic and abiotic stresses, although transgenerational inheritance of the stress-induced epigenetic changes has not been convincingly demonstrated[24–27].

Genetically dependent and independent factors contribute to epigenetic variations in plants. Genome-wide DNA methylation analysis of plants identified thousands of naturally occurring differentially methylated regions (NMRs) in various species (e.g., *Arabidopsis*, soybean and maize). Some of these NMRs are under strong genetic control by *cis* or *trans* factors[27]. For instance, natural variations in *Chromomethylase 2* (*CMT2*) caused CHH methylation variations in the wild population of *A. thaliana* that were associated with the climate at the sampling sites[28]. CHH methylation was also found to increase with growth temperature, and a genome-wide association study revealed a strong association between CHH methylation variation and genetic variants[29]. These results provide compelling evidence for plant adaption to local environments by genetically dependent epigenetic variations. Therefore, it is important to evaluate to what extent genetic differences contribute to epigenetic variation in environmental adaptation[5,9].

At the same time, DNA methylation features are not always linked to underlying genetic variation[27,30]. The rates of cytosine methylation conversion are higher than the genetic mutation rates, where the rate of gain ($2.56 \times 10^{-4}$) or loss ($6.30 \times 10^{-4}$) of methylation is five orders of magnitude higher than the genetic mutation rate ($7 \times 10^{-9}$)[31,32]. This indicates that methylome patterns can also be significantly shaped without the presence of genetic variations. For example, experimentally-induced, genetically independent differentially methylated regions (DMRs) with stable transgenerational inheritance (>8 generations), derived from the epiRILs (epigenetic recombination isogenic lines), act as epigenetic quantitative trait loci and account for 60–90% of the heritability of flowering time and primary root length[20,21]. Interestingly, 30% of the epiRIL-DMRs overlapped with NMRs among 138 natural accessions[20,21], suggesting that functional epigenetic variations might be involved in plant adaptive evolution independent of genetic variation. However, detailed functional characterization of such naturally occurring epialleles with a role in environmental adaptation has been limited.

Here, we identify a naturally occurring epiallele, NMR19-4, that controls leaf senescence by regulating the expression of *PPH* in various accessions of *A. thaliana*. Genetic analysis reveal that DNA methylation differences at NMR19-4 are independent of genetic variation. Inheritance of the methylated NMR19-4 epiallele does not depend on the RNA-dependent DNA methylation (RdDM) pathway, but the mutation in a chromatin remodeler DDM1 may change methylated NMR19-4 epiallele to an unmethylated one. The DNA methylation patterns of NMR19-4 in the 137 tested *A. thaliana* accessions are highly associated with their local climates, suggesting that methylation status of NMR19-4 might confer a selective advantage in specific environments. Molecular dating analysis show that the NMR19-4 transposon was inserted in the *A. thaliana* genome around 0.37–0.98 million years ago after the divergence of *Arabidopsis lyrata* and *A. thaliana*. Our results support the hypothesis that DNA methylation changes can mediate the effects of environment on gene expression and can contribute to plant adaptation to climate changes.

## Results

**Identification of a new copy of LINE1 in the C24 accession.** We previously identified 10,581 naturally occurring DNA methylation variation regions (NMRs) between Col-0 and C24 accessions of *A. thaliana* by whole-genome bisulfite-sequencing[33]. The methylation status of seven randomly selected loci was confirmed by Chop-PCR assays, suggesting that these NMRs are indeed differentially methylated in different accessions and that whole-genome bisulfite-sequencing reliably identifies NMRs (Fig. 1a; Supplementary Fig. 1). To investigate the inheritance patterns of these NMRs, we performed reciprocal crosses between Col-0 and C24 and found that four of the loci followed Mendelian inheritance, exhibiting a 3:1 ratio of methylated (C24) to unmethylated (Col-0) alleles in F2 populations (Fig. 1b; Supplementary Data 1). A possible explanation for the 3:1 ratio is that a recessive *trans*-factor controls the methylation status of NMRs. To test this hypothesis, we performed map-based cloning. Rough mapping indicated that the linked genetic loci for NMR7 and NMR11 were located near the NMRs themselves. In contrast, NMR19 was linked to another position on the same chromosome, away from the NMR itself (Fig. 2a; Supplementary Fig. 1), suggesting that it might be controlled by a *trans*-factor.

NMR19 was highly methylated in C24 but not in the Col-0 accession. In the TAIR10 reference genome, NMR19 is located within the 3′-terminal region of a long interspersed nuclear element 1 (LINE1) retrotransposon (AT5G41835) on chromosome 5 at ~16.75 Mb position (Fig. 1a). LINE1 is a relatively low-copy number element with 1366 copies annotated in the TAIR10 reference genome. Most of these elements are truncated fragments, with only 174 copies longer than 2 kb. To elucidate the putative *trans*-factor that controlled the methylation status of NMR19, we performed fine mapping with 1116 samples. As a result, we narrowed down the linked locus to a ~150 kb region around the 4.45 Mb position on the same chromosome (Fig. 2b). However, none of the protein-coding genes within this ~150 kb interval is obviously related to DNA methylation. As a result, we explored the possibility of a new insertion of LINE1 in this region. Next-generation sequencing (NGS) analysis (see Methods) confirmed the insertion of a truncated and inverted LINE1 specifically in C24 at 4.45 Mb on chromosome 5 (Fig. 2c), which was further verified by PCR experiments (Fig. 2d). Furthermore, in C24, we did not find a full-length copy of LINE1 at the Col-0 reference site, i.e., 16.75 Mb on chromosome 5. As expected, both copies of LINE1 at 16.75 Mb and 4.45 Mb are flanked by a typical 15 bp and 7 bp target site duplication (TSD), respectively (Fig. 2e). The NMR19 at 4.45Mb, a truncated LINE1 copy in C24, lacks the ORF1 and endonuclease coding regions of LINE1 compared with the full-length LINE1 in Col-0 (Fig. 2e). For convenience, we named the NMR19 located at 16.75 Mb

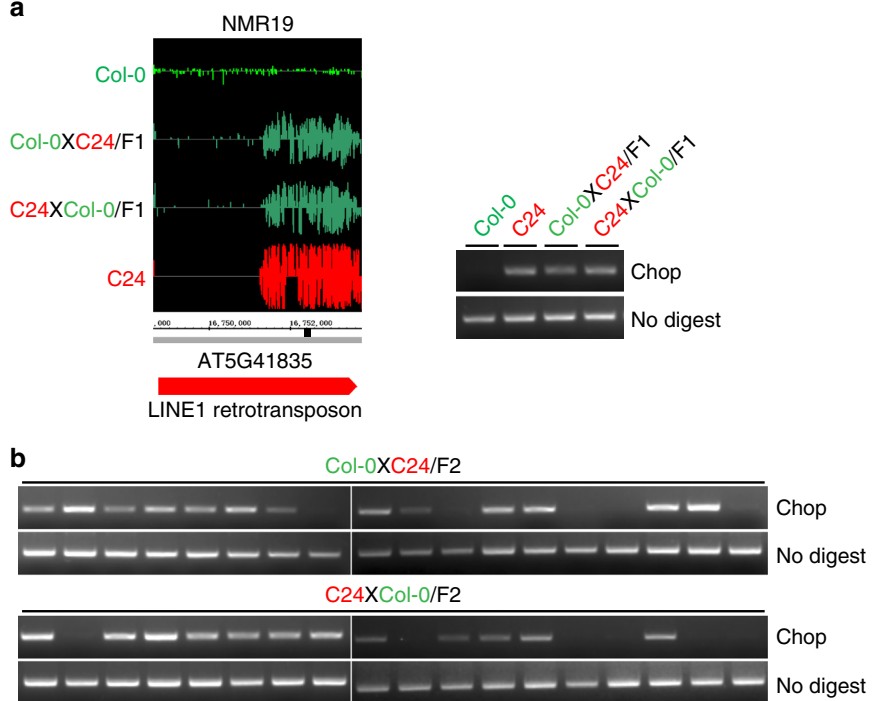

**Fig. 1** Characterization of naturally occurring DNA methylation variation region 19 (NMR19) in Col-0 and C24. **a** Integrated Genome Browser (IGB) snapshot (left) showing methylation variation between Col-0, C24 and its F1 hybrids (Col-0XC24/F1 and C24XCol-0/F1) generated from BS-seq data. Black bar below the IGB snapshot indicates the site of restriction enzyme used for DNA methylation-sensitive PCR (Chop-PCR). Gel picture (right) depicts the validation of NMR19 methylation levels by using Chop-PCR. **b** Chop-PCR analysis of methylation patterns of NMR19 in F2 population from reciprocal crosses between Col-0 and C24

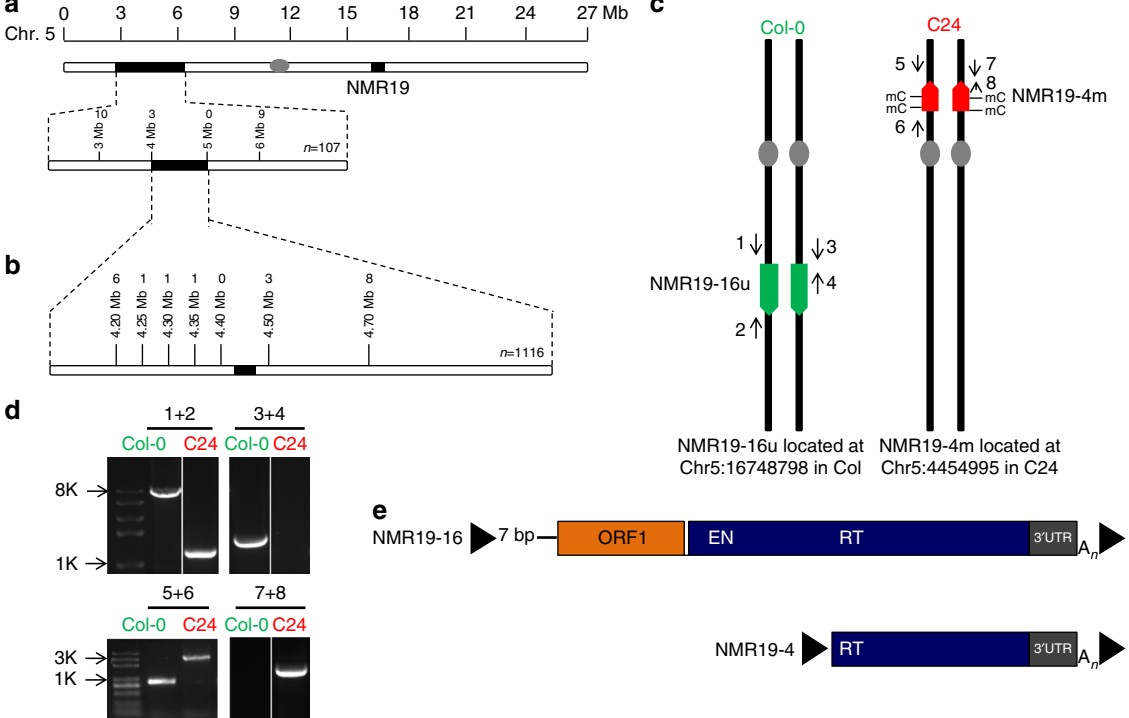

**Fig. 2** Fine mapping of NMR19 identifies a novel insertion of LINE1 in C24. **a** Rough mapping and **b** fine mapping of NMR19 methylation variation. The number of recombinants for each marker are shown above the marker. "*n*" represents the number of individuals used for the mapping. **c** Schematic representation of NMR19 insertion positions and methylation status in Col-0 and C24. mC indicates methylated NMR19 in C24. Numbered arrows indicate the positions of the primers used in **d**. **d** Products of NMR19 amplification with flanking primers. Primer positions are indicated in **c**. **e** The structure of NMR19 in Col-0 and C24 showing the positions of ORF (open reading frame), EN (endonuclease), RT (reverse transcriptase), UTR (untranslated region), and A$_n$ (polyA tail). Black triangles indicate TSD (target site duplication)

without DNA methylation "NMR19-16u" (unmethylated), and the NMR19 located at 4.45 Mb with DNA methylation "NMR19-4m" (methylated). Thus, we identified a new copy of LINE1 in C24 and found that the variation of NMR19 location, rather than a *trans*-factor, was responsible for the inheritance pattern of NMR19 methylation in the cross between Col-0 and C24.

**DNA methylation of NMR19 is independent of genetic variation.** Our analysis of NMR19 demonstrated that the position and methylation of NMR19 vary between the Col-0 and C24 accessions. To investigate the diversity of NMR19 in different accessions, we examined structural variations and DNA methylation patterns of NMR19 in 140 accessions of *A. thaliana*. These accessions came from across the globe and were obtained from the *Arabidopsis* Biological Resource Center (ABRC) (Supplementary Data 2). We classified the 140 accessions into five categories based on the genomic position and DNA methylation pattern of NMR19 (Fig. 3; Supplementary Fig. 2): (1) 78 accessions displayed neither NMR19-4 nor NMR19-16; (2) 20 accessions were like C24 and displayed only methylated NMR19-4 (NMR19-4m); (3) 5 accessions were like Col-0 and displayed only unmethylated NMR19-16 (NMR19-16u); (4) 36 accessions displayed only unmethylated NMR19-4 (NMR19-4u); and (5) 1 accession displayed both methylated NMR19-4 and methylated NMR19-16 (NMR19-4m/16m). Notably, NRM19-16 and NMR19-4, if present, are located at the same genomic context in the genomes of all the accessions examined (Supplementary Fig. 3).

To determine whether genetic variations are responsible for the DNA methylation differences of NMR19, we examined siRNA levels, NMR19 copy numbers, single nucleotide polymorphisms (SNPs) in NMR19 sequences, and SNPs throughout the whole genomes in all 140 accessions. Our results suggested that NMR19 DNA methylation is independent of genetic variation, siRNA levels and copy number; we did not observe a correlation between DNA methylation patterns and genetic variation in the genome (Supplementary Fig. 4a–c; Supplementary Figs. 5 and 6). Further, our analysis also revealed no significant correlation between *CMT2* alleles and the methylation status of NMR19 (Supplementary Fig. 7).

At some loci in *Arabidopsis* hybrids, genetic variations between the two parental alleles can be associated with allelic methylation interactions[33]. Therefore, we crossed *A. thaliana* accessions harboring methylated NMR19 with accessions that contain unmethylated NMR19, and subsequently examined NMR19 methylation levels in individual progeny from reciprocal crosses. We selected C24, Fr-2, Per-1, Nok-3, Sei-0, and Rubezhnoe-1 as methylated NMR19-4 parents, and Kro-0, Gu-0, Fi-0, and Zh as

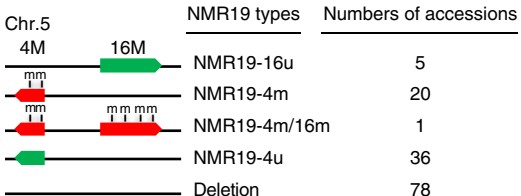

**Fig. 3** Diversity in the methylation status and position of NMR19 in different *Arabidopsis thaliana* accessions. Classification of NMR19 according to their positions and methylation status. Left, black line represents chromosome 5 and "4M" and "16M" indicate the positions of NMR19 on this chromosome, and "m" indicates methylation of NMR19. Thus, there are five different NMR19 types: NMR19-16u, NMR19-4m, NMR19-4m/16 m, NMR19-4u, and deletion. Right column shows the number of accessions with the corresponding NMR19 types

unmethylated NMR19-4 parents (Supplementary Fig. 8b). Both NMR19-4m and NMR19-4u alleles were stably inherited without *trans*-chromosomal DNA methylation or demethylation (TCM or TCdM) in F1 (Supplementary Fig. 8a) and F2 populations (Supplementary Figs. 9 and 6a). In addition, NMR19-16u and NMR19-16m showed similar patterns of inheritance as NMR19-4 (Supplementary Fig. 8a, b). Further, we confirmed the stable inheritance of the NMR19 methylation status in F1 and F2 progeny derived from reciprocal crosses between Pu2-23 and either one of the two accessions without NMR19 (Ws and Ler) (Supplementary Fig. 10). Taken together, our results show that the methylation status of NMR19 is stably inherited in both processes of hybridization and selfing.

Next, we determined whether specific histone modifications are associated with the DNA methylation status of NMR19. We found that H3K9me2, which marks heterochromatin, was enriched at NMR19-4m (C24) and NMR19-16m/4 m (Pu2-23), but not at NMR19-4u (Zh) or NMR19-16u (Col-0) (Supplementary Fig. 4d). In contrast, H3K4me3, an active chromatin mark, was enriched in NMR19-4u (Zh) and NMR19-16u (Col-0), but not at NMR19-4m (C24) or NMR19-16m/4 m (Pu2-23) (Supplementary Fig. 4d). These results show that histone modifications at NMR19 correlate with its DNA methylation status.

**Epimutation of NMR19-4 in *ddm1*-mutant background.** To delineate the mechanisms that regulate DNA methylation at NMR19, we examined NMR19 methylation in different mutant backgrounds. As mentioned above, the siRNA levels were similar among Col-0 (MMR19-16u), Zh (MMR19-4u), C24 (MMR19-4m) and Pu2-23 (NMR19-4m/16 m), suggesting that siRNAs and RdDM are not involved in the regulation of NMR19 methylation. To further test the role of RdDM, we examined DNA methylation of NMR19-4m in the RdDM mutants *nprd1*, *npre1* and *nrpd1 nrpe1*, in the C24 background[33]. We found that DNA methylation of NMR19-4 was not affected in these mutants (Fig. 4a). Furthermore, BS-seq data indicated that CHH methylation in NMR19-4 was not significantly affected in *nrpd1 nrpe1* double mutants ($p > 0.05$, Fisher exact test) (Supplementary Fig. 10), supporting our conclusion that the RdDM pathway does not regulate NMR19 DNA methylation (Supplementary Fig. 11).

We previously isolated two *ddm1* null alleles, *ddm1-14* and *ddm1-15*, in the C24 background from a screen for DNA methylation factors in *A. thaliana*. Intriguingly, some C24 plants with these *ddm1* null alleles lost DNA methylation at NMR19-4m (Fig. 4a). We also observed the same phenomenon in a published *ddm1*-mutant allele, *ddm1-9*[34]. Inbred *ddm1* plants also confirmed that *ddm1* induced stochastic methylation patterns at NMR19-4m (Supplementary Fig. 12). In contrast, *nrpd1* and *nrpe1* null alleles isolated from the screen maintained DNA methylation at NMR19. To confirm the cause-and-effect relationship between *ddm1* mutation and the loss of NMR19-4 methylation, we backcrossed *ddm1-15* that contained unmethylated NMR19-4 to wild type C24, and then examined the DNA methylation patterns of NMR19-4 in the F1, and selfed F2 and F3 progenies. The F1 progenies displayed half the methylation level of NMR19-4 (Supplementary Fig. 13a). In contrast, the F2 population contained both methylated and unmethylated NMR19-4 in either WT or *ddm1-15* homozygous backgrounds (Fig. 4b). The DNA methylation patterns of NMR19-4 of F2 WT individuals were maintained in the F3 generation (Fig. 4c; Supplementary Fig. 13b, c). Thus, although some *ddm1*-mutant plants showed unmethylated NMR19-4, the *ddm1* mutation does not necessarily lead to a loss of NMR19-4 methylation. DNA methylation of NMR19-4 may also be gradually lost in the *ddm1* background after generations. Our results suggest that loss of

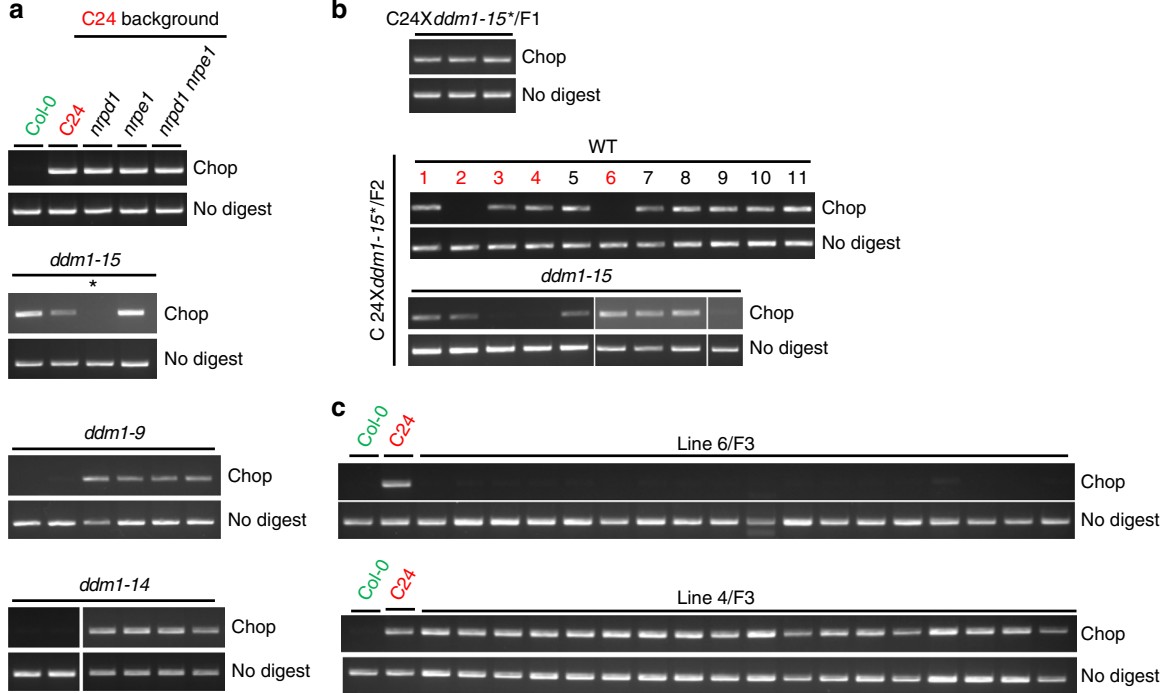

**Fig. 4** Stochastic and unrecoverable loss of methylation of NMR19-4 in *ddm1*-mutant plants. **a** DNA methylation of NMR19 in different mutants in C24 background. Asterisk denotes that the individual plant was used for backcross with C24 in the following analysis. **b** Methylation patterns of NMR19-4 in F1, F2 progenies of backcross. WT, the F2 individual plant without *ddm1* mutation; *ddm1-15*, F2 individual plant with homozygous *ddm1-15* mutation; the red colored numbers indicate the offspring of these lines used for F3 analysis in **c** and (Supplementary Fig. 9b). **c** Methylation status of NMR19-4 in F2 is maintained in F3 generation

methylation of NMR19-4 in C24 accession is irreversible, stably inherited, and thus generates an epimutation.

**NMR19-4 is a natural epiallele controlling leaf senescence.** PCR-Sanger sequencing of all the *A. thaliana* accessions containing NMR19-4 revealed that NMR19-4 was inserted in the putative promoter regions of AT5G13800 and AT5G13810, two protein-coding genes with opposite directions of transcription (Fig. 5a). To test whether NMR19-4 may be a functional epiallele that regulates gene expression, we examined the effects of NMR19-4m on the transcript levels of AT5G13800 and AT5G13810. By quantifying the transcript levels of the two genes in different *A. thaliana* accessions, we found that DNA methylation of NMR19-4 is associated with repression of AT5G13800, but not AT5G13810 (Fig. 5b; Supplementary Fig. 14). The expression levels of AT5G13800 in NMR19-4u accessions were comparable to those accessions that do not harbor NMR19 ($p > 0.05$, *t*-test two sides, Fig. 5b); and the expression levels of AT5G13800 in NMR19-4m accessions were significantly lower than those in either NMR19-4u accessions or accessions without NMR19-4 ($p < 0.01$, two-tailed *t*-test, Fig. 5b). Thus, DNA methylation of NMR19-4, rather than the presence of NMR19-4, correlated with repressed AT5G13800 expression.

AT5G13800 encodes a pheophytin pheophorbide hydrolase (*PPH*), an enzyme vital for leaf senescence via the degradation of chlorophylls[35]. Because NMR19-4m negatively regulates *PPH* expression, we tested whether NMR19-4m might lead to variations in leaf senescence in different *A. thaliana* accessions by assaying dark-induced senescence. We selected 5–6 representative accessions from each type of NMR19-4 variation to examine dark-induced chlorophyll loss. Consistent with the expression patterns of *PPH*, the accessions with NMR19-4u or without NMR19-4 exhibited similar chlorophyll contents after

dark treatment ($p > 0.05$, two-tailed *t*-test), which were significantly lower than the chlorophyll contents of accessions with NMR19-4m ($p < 0.01$, two-tailed *t*-test) (Fig. 5c, d). Thus, DNA methylation of NMR19-4 is negatively associated with the expression of *PPH* and with leaf senescence.

To determine whether DNA methylation of NMR19-4 is a causative epigenetic regulator of leaf senescence in *A. thaliana*, we selected individual F1 and F2 plants from reciprocal crosses between Gu-0 (NMR19-4u) and Fr-2 (NMR19-4m) to test whether both *PPH* expression and leaf senescence co-segregate with a particular methylation status of NMR19-4. For co-segregation assays, the same individual plant was used to detect (1) the expression levels of *PPH* by allele-specific expression assays, (2) leaf senescence induced by dark, and (3) methylation status of NMR19-4 by Chop-PCR. In the F1 hybrid and F2 segregation lines, the expression levels of *PPH* from the NMR19-4u allele was always higher than those from the NMR19-4m allele (Fig. 6a, b). Consistently, the F2 plants with homozygous NMR19-4m alleles displayed significantly higher chlorophyll contents after dark treatment, compared to the F2 plants with homozygous NMR19-4u allele (Fig. 6c). These results suggest that the three factors, methylation of NMR19-4, expression level of *PPH*, and leaf senescence, are tightly coupled and their correlations appear stably heritable. Thus, these results provide further support that the methylation status of NMR19-4 is a critical regulator of leaf senescence. To confirm that leaf senescence is epigenetically regulated by NMR19-4 methylation, we also examined the expression levels of *PPH* as well as dark-induced chlorophyll contents in 3 methylated NMR19-4 and 3 unmethylated NMR19-4 samples, which were in the C24 WT background identified from F3 populations derived from a *ddm1-15* backcross to C24. We found that *PPH* expression levels and chlorophyll contents in the three methylated samples were comparable to that in the C24 parent; whereas unmethylated

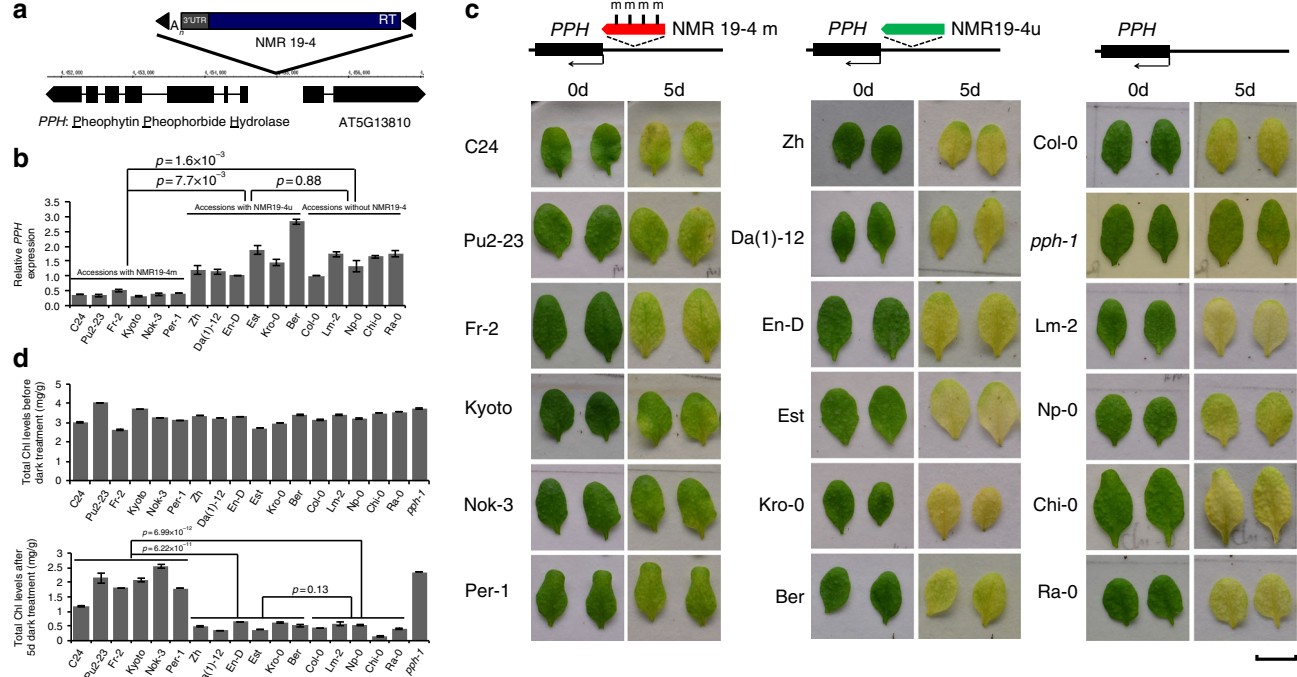

**Fig. 5** Methylation of NMR19-4 is negatively associated with leaf senescence. **a** The flanking gene structure of NMR19. black box: exon; black line: intron. Tip of the black box in gene structure depicts direction of transcription. **b** The expression of *PPH* was repressed in accessions with NMR19-4m as compared to the accessions with NMR19-4u or deleted NMR19. *p*-value was calculated by a two-tailed *t*-test, the same below. Error bars are defined as s.e.m. (*n* = 3). The expression and Chl content data of all accessions are listed in Supplementary Data 2. **c** Top schematic diagram depicts the classification of NMR19-4 into three different groups, i.e., methylated, unmethylated and deletion, and the leaf pictures below each group show darkness-induced leaf senescence phenotype in various *A. thaliana* accessions. Photographs were taken from the detached leaves before (0 d) or after incubation in darkness for 5 d, and *pph-1*, a null mutant of *PPH* in Col-0, was used as a control. Scale bars, 1 cm. **d** Quantification of chlorophyll content before and after 5 days of dark treatment in various accessions of *A. thaliana*. Chl, chlorophyll. Error bars are defined as s.e.m. (*n* = 3)

samples showed higher expression levels of *PPH* and lower levels of chlorophyll contents, compared to the methylated samples and the C24 parent (Fig. 6d–f). Taken together, these findings suggest that NMR19-4m inhibits leaf senescence by repressing the expression of *PPH*, though genetic differences likely also influence the differences in leaf senescence and *PHH* expression between NMR19-4u accessions and NMR19-4m accessions. Overall, these results suggest that NMR19-4 is a new natural epiallele that regulates leaf senescence in *A. thaliana*.

**NMR19-4 methylation status correlates with local climate**. As our analysis suggested that NMR19-4 (truncated LINE1) might have originated from the retrotransposition of NMR19-16 (full-length LINE1) (see Supplementary Notes: Origin of NMR19 elements), we asked when NMR19-4 was inserted into the 4.45Mb position of chromosome 5 in the *A. thaliana* genome. Using the NMR19-4 nucleotide polymorphism data, we performed molecular dating analysis of the NMR19-4 insertion by following the methods of Thomson et al.[36] and Studer et al.[37]. We found that NMR19-4 emerged at ~0.37–0.98 MYA (million years ago) after the separation of *A. lyrata* and *A. thaliana* (~3.5–5.8 MYA). BLAST analyses confirmed that NMR19-4 is absent but NMR19-16 is present in the orthologous positions in the *A. lyrata* genome; BLAST analyses also showed that both NMR19-4 and NMR19-16 are absent in *Capsella rubella* genomes (Supplementary Fig. 15). Therefore, LINE1/NMR19-16 might have been inserted in the *Arabidopsis* genus after it diverged from *C. rubella* but before the separation of *A. lyrata* and *A. thaliana* (Fig. 7d).

Because NMR19-4 methylation negatively correlated with the expression of the *PPH* gene and leaf senescence, we asked whether differential methylation of NMR19-4 might have contributed to environmental adaptation in geographically diverse *A. thaliana* accessions. We retrieved DNA methylation data for NMR19 from the published genome-wide MethylC-sequencing data[30], and measured the levels of *PPH* gene expression and chlorophyll contents (leaf senescence after 5 d darkness) in 137 *A. thaliana* accessions (Figs. 5b–d and 7a). These accessions were classified into three groups based on the methylation status and presence/absence of NMR19-4: the methylated group (21 accessions), the unmethylated group (39), and the deletion group (77). Representative accessions in each group are shown in Fig. 5. The DNA methylation levels of NMR19-4 in all three cytosine contexts were significantly higher in the methylated group than in both the unmethylated and deletion groups (the methylation was set to 0 if NMR19-4 is absent) (Fig. 7a). Consistent with the genetic analysis data shown above, the methylated group showed the lowest expression level of *PPH* and the highest dark-induced chlorophyll contents, i.e., the lowest degree of leaf senescence (Wilcoxon tests, *p* < 0.05, Fig. 7a). In addition, the unmethylated group was comparable to the deletion group in terms of DNA methylation status, *PPH* expression and leaf senescence (Wilcoxon tests, *p* > 0.05, Fig. 7a). Correlation analysis indicated that DNA methylation, *PPH* expression, and leaf senescence were pairwise associated with each other (Fig. 7b). These patterns are consistent with the results from our genetic analysis results, and suggest that in natural populations, DNA

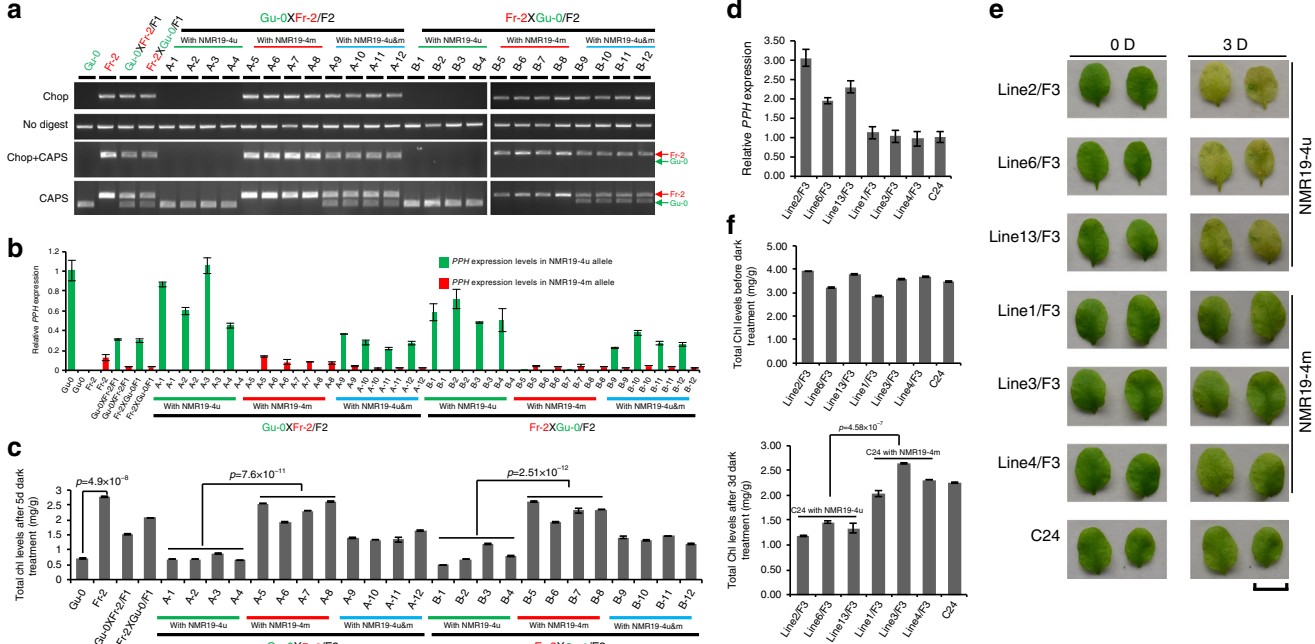

**Fig. 6** NMR19-4m controls leaf senescence. **a** Examination of methylation status of NMR19-4 based on Chop-PCR in F1 hybrids and F2 recombinant lines. The arrows at the right side of gel picture indicate the allele that corresponds to the band in the gel. The numbers of A1 to A12 and B1 to B12 in **a**–**c** indicate different individual plants and each of them was used for three different assays including Chop-PCR, allele-specific qPCR and darkness-induced leaf senescence. **b** Examination of expression levels of *PPH* in NMR19-4m and NMR19-4u alleles by allele-specific qPCR. Error bars are defined as s.e.m ($n = 3$). **c** Quantification of chlorophyll content after 5 days of dark treatment in F1 hybrids and F2 recombinant lines. *p*-value was calculated by a two-tailed *t*-test, the same below. Error bars are defined as s.e.m ($n = 3$). **d** NMR19-4m represses the expression of *PPH* compared with NMR19-4u in C24 WT background. The different lines correspond to those in Fig. 4c and Supplementary Fig. 9, the same below. We confirmed (Fig. 4; Supplementary Fig. 9) that lines 2, 6, and 13 had unmethylated NMR19-4, and lines 1, 3, and 4 had methylated NMR19-4 in C24 WT background. Error bars are defined as s.e.m. ($n = 3$). **e** NMR19-4m delays leaf senescence under dark treatment. Scale bars, 1 cm. **f** Quantification of the phenotype from **e**. Total chlorophyll levels were determined before (upper) and after (bottom) 3 d dark treatment. Error bars are defined as s.e.m. ($n = 3$)

methylation of NMR19, rather than its sequence variation, controls leaf senescence by regulating *PPH* expression.

To examine whether DNA methylation of NMR19 as well as *PPH* expression and leaf senescence has a role in environmental adaptation, we collected 19 climate parameters at two historical time points, i.e. last interglacial (LIG, 0.12–0.14 MYA) and present, at the place of origin for these *A. thaliana* accessions from WorldClim (http://www.worldclim.org/bioclim)[38]. We calculated the association of the 19 climate parameters with the three NMR19-4 related phenotypes (DNA methylation, *PPH* expression and leaf senescence) in a pairwise manner. We found that DNA methylation of NMR19-4 (including total methylation and methylation in all three sequence contexts) was associated with 7 climate parameters in the present time and 5 climate parameters in the last interglacial time (Supplementary Data 3), but *PPH* expression and leaf senescence did not show significant association with any climate parameters. Thus, DNA methylation at NMR19-4 correlates with environmental adaptation of *Arabidopsis*. These data suggest that methylation changes at NMR19-4 might facilitate environmental adaptation, perhaps not solely via regulation of *PPH* expression and leaf senescence. Among the climate parameters, the bio9 (mean temperature of Driest quarter) showed the most significant (negative) correlation with DNA methylation of NMR19-4; and the absolute correlation coefficient was higher in present than in LIG (Fig. 7b). Moreover, both the NMR19-4 deletion group and the NMR19-4u group showed significantly (Wilcoxon tests, $p < 0.05$) higher bio9 values in present than in LIG; in contrast, the NMR19-4m group showed similar (Wilcoxon tests, $p > 0.1$) bio9 values in LIG and present (Fig. 7c). Together, these results suggested that DNA methylation

levels of NMR19 may have decreased as the temperature in the driest quarter increased from LIG to present.

## Discussion

Identifying natural epigenetic variation and elucidating its role in plant adaptation is important to enhance our understanding of the epigenetic basis of biological diversity. Recent studies highlighted the role of DNA methylation variation in phenotypic responses and plant stress adaptation[3–10]. Similarly, DNA methylation patterns are strongly associated with the local geographical conditions in global accessions of *A. thaliana*, suggesting a role for epigenomic variation in adaptive evolution[11]. However, detailed insight into naturally occurring DNA methylation variation regions (NMRs) was lacking. In this context, we investigated whether genetic variation contributes to NMRs, whether NMRs are stably inherited, the mechanisms underlying the origin of NMRs, and finally whether NMRs regulate gene expression and plant adaptive traits. We identified NMR19-4 through genetic analysis of the NMRs between Col-0 and C24, and found that it is a functional natural epiallele that regulates leaf senescence. Our findings suggest that epigenetic variation might contribute to plant adaptation to their local environment.

Our genetic analysis failed to uncover any link between genetic variations and DNA methylation patterns in the examined NMRs, including NMR19. These findings are consistent with previous studies indicating that most of the NMRs are independent of genetic variations[27,30,39,40].

NMR19-4 is located within the putative promoters of two protein-coding genes with opposite directions of transcription.

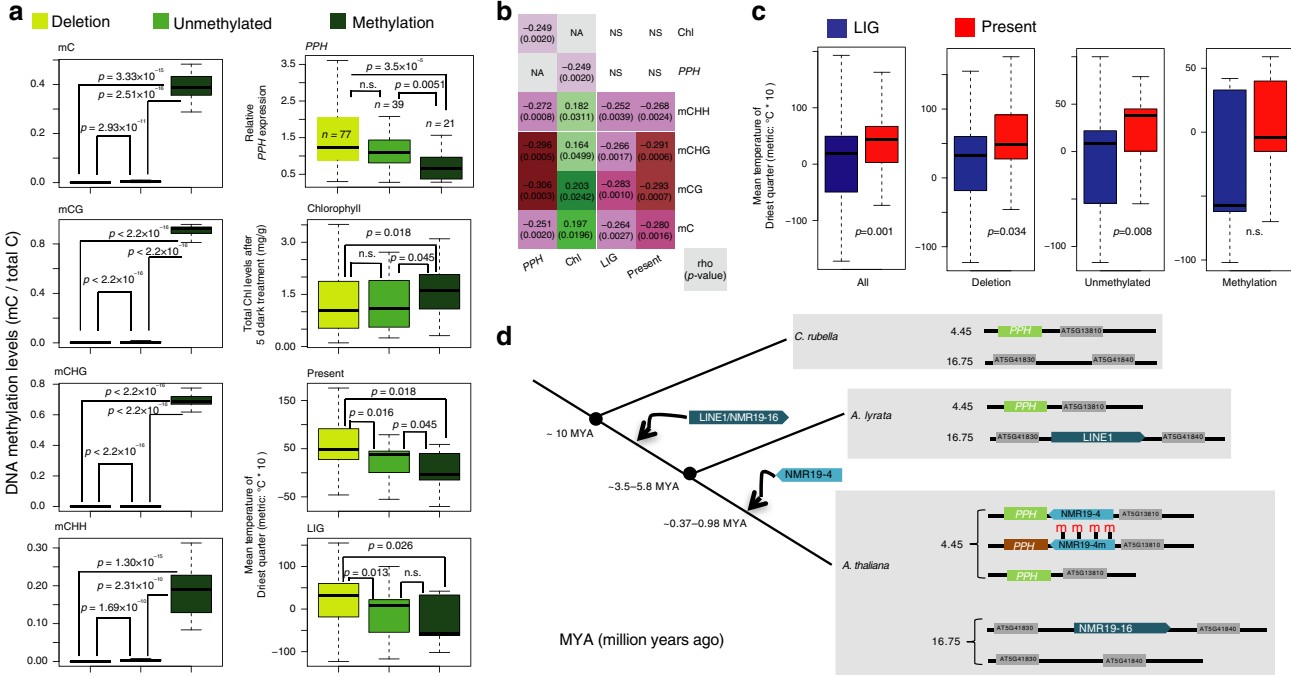

**Fig. 7** Association of DNA methylation of NMR19-4 with climate parameters at place of origin and analysis of NMR19 insertion. **a** Box-plots show methylation levels of cytosine in various sequence contexts (left), *PPH* gene expression, and chlorophyll contents, mean temperature of Driest quarter in three groups of accessions classified by methylation status of NMR19-4. Left, methylation level was calculated from the published BS-seq data[39]. Right, from upper to bottom: expression level of *PPH*, chlorophyll content, mean temperature of Driest quarter in present and last interglacial (LIG, about 120,000–140,000 years ago). *p*-value was calculated by Wilcoxon test. The box represents the distance between the 1st and 3rd quartiles and the whiskers are 1.5*IQR where IQR = Q3–Q1 (the same as below). **b** Pairwise Spearman correlation coefficients of NMR19 methylation level, *PPH* expression, chlorophyll content, temperature of Driest quarter in present and LIG. The numbers indicate correlation coefficients (rho) and *p*-values. Correlations to other climate parameters are listed in Supplementary Data. **c** The change of mean temperature of Driest quarter between present and LIG at the place of origin for accessions in different groups. *p*-value was calculated by Wilcoxon tests. **d** The pattern of NMR19 in *Capsella rubella*, *A.thaliana*, *Arabidopsis lyrata* (modified from published paper[62]). NMR19-4 is unique to *A.thaliana* and originated from a retrotransposition event that occurred 0.37–0.98 MYA. 4.45 and 16.75 represent two genomic locations of NMR19 on chromosome 5

We found that the expression of *PPH*, located downstream of NMR19-4, was affected by the methylation status of NMR19-4. Our analysis of 137 *A. thaliana* accessions, along with *ddm1*-mutants induced epimutations, showed that the DNA methylation at NMR19-4 negatively regulates the expression of *PPH* and chlorophyll degradation.

Finally, we uncovered significant associations of 12 climate parameters with NMR19-4 methylation patterns, suggesting that DNA methylation of NMR19-4 might have had a role in the local adaptation of *Arabidopsis*. Among the tested climate parameters, the bio9 (mean temperature of Driest quarter) was best associated with NMR19-4 methylation. Intriguingly, the absolute coefficient value of bio9 and NMR19-4 DNA methylation levels in LIG was lower than that in present, suggesting that the change of NMR19 DNA methylation from paleoclimates to present is potentially an adaptive process. The climate temperature of the local environment increased from LIG to present in NMR19-4 deleted and unmethylated groups, but did not change significantly in the methylated NMR19-4 groups. Molecular dating analysis indicated that NMR19-4 (originating from NMR19-16) was inserted in the *A. thaliana* genome after the separation of *A. lyrata* and *A. thaliana*. It is possible that NMR19-4 was highly methylated in all *A. thaliana* accessions when it was inserted in the genomes ~0.37–0.98 MYA; then NMR19-4 was subjected to demethylation and deletion with the increase of local temperature in hot and dry seasons, leading to diversity in the methylation and genomic location of NMR19 in *A. thaliana* wild populations. It is tempting to speculate that with global warming NMR19-4 will eventually

disappear in *A. thaliana* accessions. Therefore, epigenetically mediated adaptations may be directional in nature. In our analysis, *PPH* expression and leaf chlorophyll content did not associate with any climate parameters. It is possible that a larger sample size may reveal a correlation between the environmental parameters and the phenotype and gene expression. It is likely that DNA methylation of NMR19-4 is not the sole factor regulating *PPH* expression and leaf senescence in *A. thaliana* accessions, since complex regulatory mechanisms are involved in leaf senescence[41]. On the other hand, NMR19-4 methylation might affect not only *PPH* expression but also the expression of other gene(s) that contribute to climate adaptation. Alternatively, the correlation between DNA methylation of NMR19-4 and climate data may not reflect an adaptive role for DNA methylation.

We found that the mean temperature of Driest quarter correlates best with the methylation status of NMR19-4. We consider two possible explanations for how accessions with NMR19-4u might display better fitness than NMR19-4m accessions in environments with a high temperature in driest quarter. Firstly, NMR19-4 demethylation triggers high expression levels of *PPH* that can accelerate chlorophyll degradation, resulting in plants with reduced photosynthesis, which need less transpiration. Consequently, NMR19-4u accessions may reduce the consumption of water, which is beneficial to plants grown in environments with high temperature of the driest quarter. Secondly, NMR19-4u can promote early leaf senescence, resulting in faster completion of the life cycle, thus avoiding the encounter of plants to hot and dry environments. These mechanisms, at least in theory, could

partly explain how NMR19-4 may be involved in adaptation of *A. thaliana* during evolution. Gugger et al.[42] found a climate-associated CG single-methylation variant at 10 kb downstream of the Staygreen gene in *Quercus lobata*, implying that leaf senescence may be a general epigenetically regulated trait important for plant adaptation to climate change.

The association of NMR19-4 methylation with climate parameters may also indicate the possibility that the methylation status can respond to environmental changes. The possibility that DNA methylation of NMR19-4m was decreased with the increase of environmental temperature prompted us to test if heat treatment may cause the demethylation of NMR19-4m. We subjected accessions with NMR19-4m to heat shock (37 °C) and also grew they in ambient high temperature (30 °C), but failed to find any significant changes in DNA methylation levels (Supplementary Fig. 16). It is possible that the effects of environmental factors, such as temperature, on DNA methylation in field conditions are relatively long-term events. Hagmann et al.[43] also found that whole-genome DNA methylation variations were not associated with the local environment factors in a century-scale experiment on diverse *A. thaliana* lineages. The relatively long-term environmental effects on epigenetic status coincide with the requirements of local adaptation. If environment factors can affect epigenetic status within a short period, such as in one generation, the epigenetic status would fluctuate and it would be difficult for the fluctuating changes of epigenetic status to mediate the effect of long-term climate changes. In fact, the epigenetic changes caused by short-term environmental stress are not heritable in general[24,25]. Although some effects of repetitive salt treatments could be maintained due to intergenerational hyperosmotic stress memory, the acquired epigenetic and phenotypic changes were gradually reset to their original states in subsequent generations in the absence of stress[44]. Thus, long-term experiments may be necessary to detect the lasting effects of environmental stress on plant epigenetic status in multiple generations, as well as to estimate the rates at which the applied environmental conditions alter heritable epigenetic status.

If NMR19-4 methylation status responded to climate changes, it would be interesting to speculate what epigenetic factors might have been responsible for environment-dependent alterations in NMR19-4 methylation levels. Our ChIP assays showed that the DNA methylation of NMR19-4 was positively associated with H3K9me2, indicating that DNA methylation of the epiallele is related to the local chromatin state. The loss of DNA methylation of NMR19-4m in some individuals in a population of *ddm1* null alleles implies that DDM1 may somehow be involved in the induction of epialleles, as suggested previously[45]. The failure of DNA methylation maintenance in *ddm1* may cause epialleles. Altered NMR19-4 methylation might be driven by spontaneous epimutational events[46]. Transient genetic mutations or inactivation (for example, environmental factors that induce a transiently inactive state in a *DDM1*-like gene) could have caused the stochastic change in NMR19-4 methylation during the evolution of *A. thaliana*.

Our study not only provided insights into the involvement of epigenetic regulation in evolution and local adaptation, but also suggest the contribution of TE movement to natural phenotypic variation and organismal evolution. The structural and epigenetic variations represented by NMR19 are difficult to identify by simply re-sequencing different *A. thaliana* accessions, suggesting that the roles for TE-induced epigenetic variation in phenotypic plasticity and environmental adaptation have been underestimated[47,48]. Although genetic mutations have a primary role in natural selection and evolution, epigenetically mediated adaptation has a complementary role. More studies are needed to illuminate the role of epialleles in plant adaptation to the environment and to elucidate the underlying molecular mechanisms.

## Methods

**Plant materials and growth conditions.** All plants were grown under a long-day condition (16 h light/8 h dark). For seedling growth, *Arabidopsis* seeds were plated on 1/2-Murashige-Skoog (MS) medium with 0.6% agar and 1.5% sucrose and stratified for 7 days at 4 °C in darkness before being transferred to the growth chamber (16 h of light and 8 h of darkness, 22 °C). All mutant lines used in this study were in the C24 background as described previously[49], except for *pph-1* (SALK_000095), which was in the Col-0 background[35]. The *A. thaliana* accessions referred here are the same as those used in a previous study[30], and were ordered from *Arabidopsis* Biological Resource Center (http://www.arabidopsis.org).

**Published genomic data.** The BS-seq data of NMR19 for all the studied accessions were derived from a previous study[30]. The other NGS data were derived from NCBI GSE72993[33].

**PCR assay.** The CTAB method was used for all DNA extraction. For Chop-PCR, 1 µg of genomic DNA was digested with methylation-sensitive enzymes *Msp*I or *Hha*I overnight in a 50 µl reaction mixture. After digestion, PCR was performed using 2 µl of the digested DNA as template in a 20 µl reaction mixture. Next, the cleaved amplified polymorphic sequence (CAPS) was performed to determine the methylation status of different alleles. Primers used in Chop-PCR and CAPS are listed in Supplementary Data 4. In addition, all uncropped gels are presented in Supplementary Figs. 17–20.

**Mapping of NMR19.** On the basis of published BS-seq data, a series of Chop-PCR markers were developed for detecting the methylation status of indicated regions. The methylation status of NMR19 was used as a phenotype for mapping. The F2 population used in mapping was derived from a Col-0 and C24 cross, and genomic DNA was extracted from 2-week-old seedling of F2. Then, the methylation level of NMR19 of individual plant was detected by Chop-PCR. At last, 1223 plants with unmethylated NMR19 in F2 generation were selected for linkage analysis.

**Real-time quantitative RT-PCR.** For real-time RT-PCR analysis, total RNA was extracted from 2-week-old seedling using the TRI Reagent (Sigma) according to the manufacturer's instructions. After TURBO DNase I (Ambion) treatment, 2 µg of RNA was subject to reverse transcription reaction using the TransScript One-Step gDNA Removal and cDNA Synthesis SuperMix kit (TransGen Biotech). The cDNA reaction mixture was then diluted seven times, and 5 µl was used as template in a 25 µl PCR reaction with TransStart Tip Green qPCR SuperMix (TransGen Biotech) or SYBR Premix Ex Taq (Tli RNaseH Plus) (TaKaRa). All the reactions were carried out on a CFX96™ Real-Time System (Bio-Rad). For all reactions, *ACTIN7* was used as an internal control. All reactions were performed in three biological replicates, except for the data in Fig. 6b, where three technical replicates were performed. Primers are list in Supplementary Data 4.

**Small RNA northern blot analyses.** Total RNA was extracted from *A. thaliana* flowers according to the standard protocol of TRI Reagent (Sigma). The small RNA fraction was precipitated using the PEG method. In brief, an equal volume of PEG 8000 solution (20% PEG 8000, 1 M NaCl) was added to the total RNA. After centrifugation at 16,000×*g* for 15 min at 4 °C, and 2.5 volume of ethanol, 0.1 volume of 3 M NaAC and 1 µl glycogen were added to the supernatant. The resulted mixture was incubated at −20 °C for overnight and then centrifuged at 16,000×*g* for 15 min at 4 °C, after which the small RNA pellet was cleaned, dried, re-suspended in DEPC-treated water. For each sample, 10 µg of small RNA was separated on a 15% polyacrylamide gel, which was electrotransferred to a Hybond N + membrane (GE Healthcare Life Sciences). Membranes were cross-linked, incubated for 2 h at 80 °C, and hybridized overnight at 38 °C with $^{32}$P-labeled DNA probes or oligonucleotides (listed in Supplementary Data 4) in PerfectHyb buffer (Sigma). Washed membranes were exposed to X-ray films at -80 °C for 3 days. In addition, the uncropped blots are presented in Supplementary Figure 18.

**ChIP assays.** Overall, 3 g 2-week-old seedling was mixed with PBS buffer (0.01 M NaH$_2$PO$_4$, 0.01 M Na$_2$HPO$_4$, pH 7.0) with 1% formaldehyde, cross-linked by vacuum infiltration for 15 min changing vacuum two times (10 min–5 min), and the crosslink was stopped using 1/15 volume of 2 M glycine by vacuum infiltration for 5 min. After decanting the solution, the plant tissues were dried with kimwipes, frozen in liquid nitrogen and ground to powder. Powder of tissues was re-suspended in 30 ml of HB buffer (2.5% Ficoll 400, 5% Dextran T40, 0.4 M Sucrose, 25 mM Tris pH 7.4, 10 mM MgCl$_2$, 0.035% β-mercaptoethanol, 1% Protease Inhibitor Cocktail (Sigma)), homogenized and filtered through Miracloth (Millipore). Triton x-100 was added to the supernatant until final concentration was 0.5%. After spinning at 2000×*g* for 20 min at 4 °C, the pellet was re-suspended in HB buffer containing 0.1% Triton x-100 and spun 2000×*g* for 10 min at 4 °C. Isolated nuclei were washed in GB buffer (0.5 M hexylene glycol, 5 mM PIPES-

KOH pH 7.0, 10 mM $MgCl_2$, 1% Triton x-100, 0.0175% β-mercaptoethanol, 10 μM MG132, 0.1% Protease Inhibitor Cocktail (Sigma), 1 mM Benzamidine, 0.2 mM PMSF). After spinning at 3000 rpm for 10 min at 4 °C, the pellet was re-suspended in 500 μl of Nuclei Lysis buffer (50 mM Tris pH 8, 10 mM EDTA, 1% SDS, 1 mM PMSF, 5 mM Benzamidine, 50 μM MG132, 1% Protease Inhibitor Cocktail (Sigma)). Bioruptor$^{TM}$ UCD-200 sonicator (diagenode) was used to sonicate chromatin with a 15 s pulses of high power and 30 s cooling between pulses for 30 min. Following centrifuge at 21,130×g for 5 min at 4 °C, 1/5 and 4/5 of supernatant were used for INPUT total DNA control and immunoprecipitation, respectively. After adding 9 volume of ChIP dilution buffer (1.1% Triton x-100, 1.2 mM EDTA, 16.7 mM Tris pH 8.0, 167 mM NaCl, 0.1% PMSF, 1% Protease Inhibitor Cocktail (Sigma)) to supernatant, it was pre-cleared with 10 μl of Dynabeads Protein G (Invirogen) for 1 h at 4 °C. After removing the beads from mixture, the supernatant was incubated with the appropriate antibody (5 μl of anti-H3K4me3 antibody (Millipore, #04-745), 10 μl of anti-H3K9me2 antibody (Abcam, #ab1220)) for overnight at 4 °C. Next, after adding 20 μl of Dynabeads Protein G, the mixture was incubated for 2 h at 4 °C. Beads were sequentially washed with 1 ml of the following buffers: Low Salt Wash buffer (150 mM NaCl, 0.1% SDS, 1% Triton x-100, 2 mM EDTA, 20 mM Tris pH 8.0), High Salt Wash buffer (500 mM NaCl, 0.1% SDS, 1% Triton x-100, 2 mM EDTA, 20 mM Tris pH 8.0), LiCl wash buffer (250 mM LiCl, 1% Igepal, 1% Sodium Deoxycholate, 1 mM EDTA, 10 mM Tris pH 8.0), TE buffer (10 mM Tris pH 8.0, 1 mM EDTA). Immunocomplexes were eluted with 250 μl of Elution buffer (1% SDS, 0.1 M $NaHCO_3$) at 65 °C for 15 min. After reverse crosslink, 10 μl of 0.5 M EDTA, 20 μl of 1 M Tris pH 6.5 and 1 μl of proteinase K (Invitrogen) were added to each sample, which was incubated at 45 °C for 2 h. DNA was then purified using conventional phenol/chloroform extraction and ethanol/salt precipitation. The products were eluted into 200 μl of dd-$H_2O$, from which 5 μl was used for each qPCR reaction.

**Measurement of chlorophyll content**. Chlorophyll pigments were extracted with 80% ice-cold acetone from leaf tissues of the plants by following the method of Ni et al.[50] using the Varioskan Flash spectrophotometer (Thermo Scientific).

**Darkness-induced leaf senescence assay**. The 11-day-old seedlings were transplanted to soil in greenhouse. After growth for 11 days, 5th or 6th of rosette leaves were detached and placed on plastic square PetriDishes containing three-layer filter papers at the bottom immersed in deionized water, and the plates were packaged by aluminum foil, and incubated in growth chamber for indicated periods.

**Co-segregation assay**. The 11-day-old seedlings were transplanted to soil in greenhouse. After growth for 11 days, 5th or 6th of rosette leaves from each individual plant were used for allele-specific qPCR and darkness-induce leaf senescence assay. A randomly selected rosette leave was used to detect the methylation status of NMR19-4 by Chop-PCR. The primers for allele-specific qPCR were designed by following those in published papers[51–53]. Primers are list in Supplementary Data 4.

**Detection of NMR19-4 insertions in C24 genome**. Detection of new NMR19 insertions in C24 genome was performed according to previous report[54]. Briefly, 150 bp paired-end sequencing library was constructed and sequenced by an Illumina HiSeq 2500 sequencer. To detect possible NMR19 insertions in the C24 genome, we analyzed the read pairs with one end being mapped to NMR19-16 sequence, and the other end being mapped to unrelated sequences in the genome. The mapping location of an unrelated sequence, if uniquely mapped in the genome, may be used to locate a potential insertion of NMR19.

**Phylogenetic analyses of accessions and NMR19 sequences**. SNP data of 137 *Arabidopsis* accessions were downloaded from the 1001 genome project (http://1001genomes.org/), and used to construct the phylogenetic tree of the *A. thaliana* accessions using the neighbor-joining (NJ) method in MEGA7 with default parameters[55]. To construct the phylogenetic tree of NMR19 sequences, we first sequenced the NMR19 regions in the selected accessions, aligned the sequences using MUSCLE (MUltiple Sequence Comparison by Log-Expectation)[56] and finally constructed the NMR19 tree using the NJ method in MEGA7 with default parameters[55]. The sequence of NMR19 was listed in Supplementary Data 5.

**Estimation of the insertion time of NMR19-4**. On the basis of Eq. (1) of Thomson et al.[36], we used variations between NMR19 orthologous sequences from *A. thaliana* (Col-0) and *A. lyrata* and calculated the substitution rate per site per year as $6.8 × 10^{-9}$, which is roughly equal to $7 × 10^{-9}$ that was calculated using number of spontaneous mutations that were accumulated in 30 generations in a single-seed descent *Arabidopsis* population[31] and falls within the range of $5–30 × 10^{-9}$[57] in plant nuclear genomes. Subsequently, we aligned NMR19-4 nucleotide sequences from 56 *A. thaliana* accessions using MUSCLE[58] and manually adjusted the alignments. Assuming a star phylogeny and following the method of Studer et al.[37], we obtained the time of NMR19-4 insertion to be ~0.37 MYA. Given that this calculation is based on star phylogeny, which may lead to underestimation of

insertion time when two sequences are from recently diverged accessions. In addition, NMR19-4 may be under natural selection, which would also lead to the underestimation of insertion time. Furthermore, we employed another method by estimating the divergence time from the most recent common ancestor (MRCA) of NMR19-4. In brief, we inferred the ancestor sequence using the NMR19-4 sequences from different *Arabidopsis* accessions using the maximum likelihood method[59] under the Tamura-Nei model[60], and then counted the substitutions between each NMR19-4 sequence and the ancestor sequence, integrated the numbers with substitution rate into the equation 3[36], and finally estimated the insertion time to be ~0.98 MYA. Note that the estimate based on MRCA is usually an overestimate[37] and thus represents an upper bound. Taken together, we concluded that NMR19-4 was inserted into the *A. thaliana* genome from 0.37 to 0.98 MYA.

**Statistical analysis**. Statistical analyses were performed using R packages[61].

**Data availability**. All data supporting the finding of this study are available within the manuscript and its supplementary files or are available from the corresponding author upon request.

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

## Acknowledgements

We thank Dr. Ya-Long Guo and Mr. Tingshen Han for providing us the climates data. We thank Ms. Quanhua Chen for helping us to do darkness-induced leaf senescence assay. We thank Dr. Craig Pikaard for providing us the *ddm1-9* seeds. This work was supported by the Chinese Academy of Sciences (to J.-K.Z.).

## Author contributions

L.H., Q.Z. and J.-K.Z. designed the research. L.H. performed experiments. W.W. and Q.Z. performed the evolution and adaptation analysis. L.H., W.W., L.Y., D.W., Z.Z., H.H., R.L., H.Z. and Q.Z. analyzed the data. L.H., G.Z., Q.Z., R.L. and J.-K.Z. wrote the manuscript.

## Additional information

**Competing interests:** The authors declare no competing financial interests.

