## [Peer Review File · Nature Communications]

Reviewers' comments:

Reviewer #1 (Remarks to the Author):

Experimental studies have clearly shown that epialleles can be stably inherited and affect plant traits. It remains largely unknown to which extent epiallelic variation is independent of genetic diversity in the wild and to which extent it contributes to plant phenotypes in natural settings. The major bottleneck is the difficulty to confidently identify epialleles in diverse genetic backgrounds and in partly incomplete genome assemblies (as shown in this study).

This is a careful and important study that identifies a naturally occurring epiallele for leaf senescence. The mythological approach that led to the identification and characterization of this epialleles was not trivial and I congratulate the authors on their work. I recommend publication of this article.

Here are a few suggestions:

Why introduce the term NMR (naturally occurring differentially methylated region); why not use the term DMR (differentially methylated region). What is natural or not cannot be clearly defined and the distinction is actually irrelevant. In line 116, the authors mention "naturally occurring DMRs" to refer to NMRs. So, there is also some inconsistency with the notation. Please simplify and make consistent.

Line 13: "suggesting the role of environmental factors in shaping methylation patterns that ultimately lead to plant adaptation". I would be careful here. This has a non-Darwinian interpretation without empirical support. The methylation status of this locus could have also changed spontaneously and be subject to Darwinian selection in response to environmental pressures. This is different than to say that the environmental factors shape methylation patterns as if they are targeted environment-induced changes.

In the introduction or discussion, the author may want to bring in some ideas from Vidalis et al. *Genome Biology*, 2016 (Methylome evolution in plants), particularly with regards to interplay between genomic evolution (as exemplified by their LINE1 insertion) and epimutational processes as exemplified by the stochastic loss of the methylation status of that new insertion.

Line 147: The reasoning is not so clear to me here. At this point of their presentation, the data could also suggest that these loci are targets of allelic TCM events, which would explain the 3:1 ratio in the F2s (note: later in Line 220 TCM events are ruled out, but the reader does not know this yet). Why does this observation lead to identification to trans-acting factors? Perhaps one should say: "One possibility for the 3:1 ratio is that". But please note that this trans-acting hypothesis only works under the constraint that the trans-acting locus is not dominant.

Line 202: Although I believe the the methylation status of NMR19 is mainly independent of genetic variation, particularly based on subsequent experiments in this ms, the genetic diversity analysis does not seem like the right test to rule out genetic effects. I do not think that genome-wide genetic divergence could pick up situations where genetic polymorphisms at a single locus control NMR19 methylation. They would get drowned out.

In contrast to the results section of this paper, I find the Discussion too speculative in parts. For instance, I would remove sentences such as: "...our findings are consistent with the theoretical model in which the contribution of epimutations to adaptive walks is maximal when alternative epialleles have minimal fitness effects". The authors refer to a review written by two experimentalists. Unless the authors refer to the original theoretical study that shows this, this reference seems too

speculative.

On the other hand, could the authors comment more on the mechanisms that could drive stochastic loss of methylation at NMR19. That the *ddm1* mutation induces such stochastic events is interesting. Could one imagine transient mutations in *ddm1*-like genes during the evolutionary past of *A. thaliana*?

The section starting line Line 317 is the weakest part of the paper, partly owing to the lack of large samples sizes to render the ecological inferences meaningful and owing to the fact that methylation status but not phenotype and gene expression correlate with environmental parameters. This weakness takes away from an otherwise solid paper. One should really consider moving this section into the discussion part.

Reviewer #2 (Remarks to the Author):

This is a study on a phenomenon that epigenetic mutations on retrotransposons can trigger expression of adjacent genes. The concept of retrotransposon- gene regulation is far from new, but the value of this study is on the existence of naturally occurring epialleles from various *Arabidopsis* accessions, although these have also been reported previously.

From BS-seq data of Col-0 and C24 accessions, the authors identified a locus with naturally occurring DNA methylation variations, named NMR19. This NMR19 corresponded to LINE1 retrotransposon showing variations in different accessions. By genetic analyses, the authors showed methylated NMR19 copies did not affect unmethylated copies and vice versa. Either unmethylated or methylated version was stably inherited through generations. Mutations in genes involved in RNA-dependent DNA methylation (RdDM) pathways also did not change methylated NMR19.

DNA methylation levels or the existence of NMR19-4 controlled RNA levels of PHEOPHYTIN 40 PHEOPHORBIDE HYDROLASE (PPH), which resulted in leaf senescence phenotype. The phenotype data was quite clear. Levels of NMR19 DNA methylation and RNA levels of PPH and senescence phenotypes were quite well correlated each other in their datasets. Lastly, the authors claim long-term climate conditions have driven DNA methylation levels of NMR19 in *Arabidopsis* accessions.

Overall, strong parts in this study are identification of a differentially methylated region, NMR19 and its effects on PPH gene expression and senescence. Weak parts are that there are not much new finding in terms of epigenetic regulation of NMR19 and insufficient explanation about the link between PPH expression and temperature.

Major points

-Figure 4a: They could do BS-PCR and Sanger sequencing to confirm if there is any changes in CHH context by *nrpd1* and *nrpe1* mutations.

-Loss of 24nt siRNA pathway may not be sufficient to remove all contexts of DNA methylation and heterochromatin marks on NMR19-4m. One possible experiment is to express hairpin RNA construct (producing 21 and 24 nt small RNAs) in one NMR19-4u accession to confirm there is no effects of RdDM on NMR19-4.

- Fig. 4b the methylation pattern is stochastic in *ddm1* and in the backcrossed population. Inbred

ddm1 plants can be used to clarify if DDM1 is indeed associated with methylation level of NMR19. Especially when they conclude by their statement on page 14 Line 411-415.

-Page 9, Line 252: It would be good to mention that unmethylated NMR19-4 might be present in F1 but it was masked by methylated element in C24 parental line. qPCR approach might distinguish heterozygosity of NMR19-4 methylation level in F1 samples.

- Page 16: They used 30C degree to induce unmethylation of NMR19, but can they try higher temperature like 37C degree as many other groups used this condition to observe epigenetic changes?

- RdDM is regulated by DRM1/DRM2 pathway. On the other hand, CMT2 mutation status in Arabidopsis ecotype accessions was associated with adaptation in temperature (Shen et al. 2014 Plos Genet). Have the authors tried to find any correlation between CMT2 alleles and NMR19 DNA methylation levels?

-They describe too much about the roles of climate changes in methylation of NMR19 relying on publicly available data. Instead, it would be nice if they discuss about relationship between leaf senescence and temperature. For example, how higher PPH RNA expression can positively affect adaption in elevated temperature.

Minor points

- It would be nice to have a DNA sequence alignment between NMR19-16u and NMR19-4m elements with primer sequence information. Chop-PCR, small RNA blot, ChIP-PCR data rely on these sequences. They could show this alignment with representative accession form example Col, C24, Pu2-23

- Supplementary Fig 4C: There is a typo: Tatol RNA -> Total RNA

-Page 9: a typo Fig. 3b -> Fig. 4b

-‘Thus, gain or loss of methylation of NMR19-4 represents a 416 natural epimutation that is independent of genetic variation’ Lin 416-417-> I am not sure if this statement is relevant or sufficient based on their datasets.

- A typo in Supplementary Fig. 10a: label C24xrdm1 ddm1-15/F2 -> C24 x ddm1-15/F2?

Point-by-point response to reviewer comments

Reviewer #1 (Remarks to the Author):

Experimental studies have clearly shown that epialleles can be stably inherited and affect plant traits. It remains largely unknown to which extent epiallelic variation is independent of genetic diversity in the wild and to which extent it contributes to plant phenotypes in natural settings. The major bottleneck is the difficulty to confidently identify epialleles in diverse genetic backgrounds and in partly incomplete genome assemblies (as shown in this study).

This is a careful and important study that identifies a naturally occurring epiallele for leaf senescence. The mythological approach that led to the identification and characterization of this epialleles was not trivial and I congratulate the authors on their work. I recommend publication of this article.

Here are a few suggestions:

(1) Why introduce the term NMR (naturally occurring differentially methylated region); why not use the term DMR (differentially methylated region). What is natural or not cannot be clearly defined and the distinction is actually irrelevant. In line 116, the authors mention "naturally occurring DMRs" to refer to NMRs. So, there is also some inconsistency with the notation. Please simplify and make consistent.

Response: We thank the Reviewer for giving us the opportunity to clarify why we introduced the term NMRs (naturally occurring differentially methylated regions) rather than using the existing term DMRs (differentially methylated regions). DMRs have been widely used in the plant community to compare the methylation status of wild-type and mutants. To distinguish such differentially methylated regions that are produced by the dysfunction of DNA methylation machinery, from the regions which are differentially methylated in naturally occurring populations, the term NMR was introduced.

We apologize for the inconsistency with the notation, and have carefully checked and revised to make it consistent throughout the manuscript.

(2) Line 131: "suggesting the role of environmental factors in shaping methylation patterns that ultimately lead to plant adaptation". I would be careful here. This has a non-Darwinian interpretation without empirical support. The methylation status of this locus could have also changed spontaneously and be subject to Darwinian selection in response to environmental pressures. This is different than to say that the environmental factors shape methylation patterns as if they are targeted environment-induced changes.

Response: We agree with the reviewer that the methylation status of NMR19 could have changed spontaneously and was subjected to Darwinian selection. We have revised relevant statements in the Introduction and Discussion to reflect this view (e.g.

lines 114-115 and 465-466).

(3) In the introduction or discussion, the author may want to bring in some ideas from Vidalis et al. Genome Biology, 2016 (Methylome evolution in plants), particularly with regards to interplay between genomic evolution (as exemplified by their LINE1 insertion) and epimutational processes as exemplified by the stochastic loss of the methylation status of that new insertion.

Response: Thank you for the useful suggestion. Indeed, Vidalis et al. provided some important clues regarding the mechanisms that might alter NMR19-4 methylation. We have brought these ideas into the discussion section of manuscript, and suggested that "Altered NMR19-4 methylation might be driven by spontaneous epimutational events⁴⁵. Transient genetic mutations or inactivation (for example, environmental factors that induce a transiently inactive state in a *DDM1*-like gene) could have caused the stochastic change in NMR19-4 methylation during the evolution of *A. thaliana*." (Lines 465-469).

(4) Line 147: The reasoning is not so clear to me here. At this point of their presentation, the data could also suggest that these loci are targets of allelic TCM events, which would explain the 3:1 ratio in the F2s (note: later in Line 220 TCM events are ruled out, but the reader does not know this yet). Why does this observation lead to identification to trans-acting factors? Perhaps one should say: "One possibility for the 3:1 ratio is that". But please note that this trans-acting hypothesis only works under the constraint that the trans-acting locus is not dominant.

Response: Thank you for pointing this out. We have changed the sentences in Lines 133-135 to "A possible explanation for the 3:1 ratio is that a recessive *trans*-factor controls the methylation status of NMRs. To test this hypothesis, we performed map-based cloning."

(5) Line 202: Although I believe the the methylation status of NMR19 is mainly independent of genetic variation, particularly based on subsequent experiments in this ms, the genetic diversity analysis does not seem like the right test to rule out genetic effects. I do not think that genome-wide genetic divergence could pick up situations where genetic polymorphisms at a single locus control NMR19 methylation. They would get drowned out.

Response: We agree with the reviewer that genome-wide genetic divergence analysis is not a good tool to rule out genetic effects on the methylation status of NMR19. Thus, we toned-down our statement to "Our results suggested that NMR19 DNA methylation is independent of genetic variation, siRNA levels and copy number; we did not observe a correlation between DNA methylation patterns and genetic variation in the genome" in Lines 186-189.

(6) In contrast to the results section of this paper, I find the Discussion too speculative in parts. For instance, I would remove sentences such as: "...our findings are consistent with the theoretical model in which the contribution of epimutations to

adaptive walks is maximal when alternative epialleles have minimal fitness effects". The authors refer to a review written by two experimentalists. Unless the authors refer to the original theoretical study that shows this, this reference seems too speculative.

Response: Thank you for pointing this out. We have carefully checked and removed the speculative sentences from the Discussion section (Line 420).

(7) On the other hand, could the authors comment more on the mechanisms that could drive stochastic loss of methylation at NMR19. That the *ddm1* mutation induces such stochastic events is interesting. Could one imagine transient mutations in *ddm1*-like genes during the evolutionary past of *A. thaliana*?

Response: We thank the reviewer for this comment regarding the involvement of DDM1 in NMR19 methylation. As the reviewer suggested, our results showing stochastic loss of NMR19 methylation in *ddm1* mutants suggest that transient genetic mutations or inactivation (for example, environmental factors that induce a transiently inactive state in a DDM1-like gene) could have caused the stochastic change in methylation during the evolution of *A. thaliana*. We have added this speculation to the discussion (Lines 467-469).

(8) The section starting line Line 317 is the weakest part of the paper, partly owing to the lack of large samples sizes to render the ecological inferences meaningful and owing to the fact that methylation status but not phenotype and gene expression correlate with environmental parameters. This weakness takes away from an otherwise solid paper. One should really consider moving this section into the discussion part.

Response: We agree with the reviewer that a large sample size is required to make strong ecological inferences. Our conclusions are based on data from 141 accessions, including molecular evolution studies, gene expression analysis, methylation analysis, chlorophyll quantification, and correlation analyses with climate parameters. These conclusions are consistent with the results from "Epigenomic Diversity in a Global Collection of *Arabidopsis thaliana* Accessions" (Kwakatsu et al., 2016), where methylation levels within TEs were negatively correlated with temperatures in the driest and warmest quarter. We would like to keep this part in the Results section. We do appreciate the reviewer's point, though, so we have indicated in the Discussion that a larger sample size may reveal a correlation between the environmental parameters and the phenotype and gene expression (Lines 415-417).

Reviewer #2 (Remarks to the Author):

This is a study on a phenomenon that epigenetic mutations on retrotransposons can trigger expression of adjacent genes. The concept of retrotransposon- gene regulation is far from new, but the value of this study is on the existence of naturally occurring epialleles from various Arabidopsis accessions, although these have also been reported previously.

From BS-seq data of Col-0 and C24 accessions, the authors identified a locus with naturally occurring DNA methylation variations, named NMR19. This NMR19 corresponded to LINE1 retrotransposon showing variations in different accessions. By genetic analyses, the authors showed methylated NMR19 copies did not affect unmethylated copies and vice versa. Either unmethylated or methylated version was stably inherited through generations. Mutations in genes involved in RNA-dependent DNA methylation (RdDM) pathways also did not change methylated NMR19.

DNA methylation levels or the existence of NMR19-4 controlled RNA levels of PHEOPHYTIN 4O PHEOPHORBIDE HYDROLASE (PPH), which resulted in leaf senescence phenotype. The phenotype data was quite clear. Levels of NMR19 DNA methylation and RNA levels of PPH and senescence phenotypes were quite well correlated each other in their datasets. Lastly, the authors claim long-term climate conditions have driven DNA methylation levels of NMR19 in Arabidopsis accessions.

Overall, strong parts in this study are identification of a differentially methylated region, NMR19 and its effects on PPH gene expression and senescence. Weak parts are that there are not much new finding in terms of epigenetic regulation of NMR19 and insufficient explanation about the link between PPH expression and temperature.

Major points

(1) Figure 4a: They could do BS-PCR and Sanger sequencing to confirm if there is any changes in CHH context by *nrpd1* and *nrpe1* mutations.

Response: We thank the reviewer for this pertinent suggestion to check the CHH methylation levels of NMR19-4, since a major function of the RdDM pathway is *de novo* methylation and maintenance of CHH methylation. In fact, we have published BS-seq data of *nrpd1 nrpe1* double mutants in C24 background (Zhang et al., 2016 PNAS). These BS-seq data also indicated that CHH methylation in NMR19-4 was not significantly affected in *nrpd1 nrpe1* double mutants ($p > 0.05$, Fisher exact test) (new data added as Supplementary Fig. 11), providing further support for our conclusion that the RdDM pathway does not regulate NMR19 DNA methylation. We have added the above information in the revised text. (Lines 225-229)

Supplementary Figure 11 Methylation profile of NMR19-4 in wild type C24 and C24-*nrpd1 nrpe1* double mutant. Shown are IGB snapshots of NMR19-4 methylation from BS-seq data.

(2) Loss of 24nt siRNA pathway may not be sufficient to remove all contexts of DNA methylation and heterochromatin marks on NMR19-4m. One possible experiment is to express hairpin RNA construct (producing 21 and 24 nt small RNAs) in one NMR19-4u accession to confirm there is no effects of RdDM on NMR19-4.

Response: We agree with the reviewer that loss of the 24nt siRNA pathway may not be sufficient to remove all contexts of DNA methylation and heterochromatin marks on NMR19-4m. Besides the data obtained from the mutants that abolish the RdDM pathway (*nrpd1 nrpe1*, see Supplementary Fig. 10 and Figure 4a in the manuscript), we have additional small RNA Northern blot data showing that both NMR19-4m and NMR19-4u produce similar levels of 24-nt siRNAs and do not produce 21-nt siRNA. These additional data provide further support that neither 21nt-small RNAs nor 24-nt siRNAs are associated with DNA methylation of NMR19-4 (Supplementary Fig. 4c).

(3) Fig. 4b the methylation pattern is stochastic in *ddm1* and in the backcrossed population. Inbred *ddm1* plants can be used to clarify if DDM1 is indeed associated with methylation level of NMR19. Especially when they conclude by their statement on page 14 Line 411-415.

Response: We thank the reviewer for this suggestion. We have now included the data from inbred *ddm1* plants in supplementary Figure 12. These data indicate that the unmethylated status was maintained in the progenies of unmethylated samples, whereas the progenies derived from methylated samples segregated into methylated and unmethylated (as shown in Supplementary Fig. 12, below). In the revised manuscript, we added a sentence "Inbred *ddm1* plants also confirmed that *ddm1* induced stochastic methylation patterns at NMR19-4m." in Lines 234-235.

(4) Page 9, Line 252: It would be good to mention that unmethylated NMR19-4 might be present in F1 but it was masked by methylated element in C24 parental line. qPCR approach might distinguish heterozygosity of NMR19-4 methylation level in F1 samples.

Response: We thank the reviewer for suggesting this experiment. We have now performed this assay and observed that the methylation level of NMR19-4 is median in the F1 of *ddm1-15*(*) backcrossed to C24 (see below). This result indicates that the unmethylated allele of NMR19-4 from *ddm1-15*(*) might be present in F1, which is in accordance with the Reviewer's comment. We have added these data to Supplementary Figure 13a and modified the sentence to "The F1 progenies displayed half the methylation level of NMR19-4" in Line 240-241.

Supplementary Figure 13 Methylation status of NMR19-4 was maintained in F3 (related to Fig. 4b).

(a) DNA methylation levels of NMR19-4 in F1 of backcross determined by restriction enzyme digestion and q-PCR. Error bars are defined as s.e.m.

(5) Page 16: They used 30C degree to induce unmethylation of NMR19, but can they try higher temperature like 37C degree as many other groups used this condition to observe epigenetic changes?

Response: To detect the effect of long-term heat stress on the methylation of NMR19-4, we incubated *Arabidopsis thaliana* at 30°C for a whole life cycle, from seed germination to maturation of new seeds. As suggested by the reviewer, we also tried heat shock (37°C) but again failed to observe any demethylation of NMR19-4m. This information is indicated in the text. (Lines 437-439)

(6) RdDM is regulated by DRM1/DRM2 pathway. On the other hand, CMT2 mutation status in *Arabidopsis* ecotype accessions was associated with adaptation in temperature (Shen et al. 2014 Plos Genet). Have the authors tried to find any correlation between CMT2 alleles and NMR19 DNA methylation levels?

Response: We thank the reviewer for this valuable suggestion. We checked the correlation between CMT2 and NMR19 DNA methylation and added the data to supplementary Figure 7. Our analysis showed that there is no significant correlation between CMT2 alleles and the methylation status of NMR19. We have added the above sentence into Lines 189-191.

(7) They describe too much about the roles of climate changes in methylation of NMR19 relying on publicly available data. Instead, it would be nice if they discuss about relationship between leaf senescence and temperature. For example, how higher PPH RNA expression can positively affect adaption in elevated temperature.

Response: We took the reviewer's suggestion into consideration and improved the Discussion of the manuscript. We shortened the text on the potential role of climate changes in methylation of NMR19, and placed more emphasis on how higher PPH expression can positively affect adaptation at elevated temperature in the driest quarter. The Discussion now reads "We found that the Mean Temperature of Driest Quarter correlates best with the methylation status of NMR19-4. We consider two

possible explanations for how accessions with NMR19-4u might display better fitness than NMR19-4m accessions in environments with a high temperature in driest quarter. Firstly, NMR19-4 demethylation triggers high expression levels of *PPH* that can accelerate chlorophyll degradation, resulting in plant with reduced photosynthesis, which need less transpiration. Consequently, NMR19-4u accessions may reduce the consumption of water, which is beneficial to plants grown in environments with high temperature of the driest quarter. Secondly, NMR19-4u can promote early leaf senescence, resulting in faster completion of the life cycle, thus avoiding the encounter of plants to a hot and dry environments. These mechanisms, at least in theory, could explain how NMR19-4 may be involved in adaptation of *A. thaliana* during evolution.” (Lines 421-432).

Minor points

(8) It would be nice to have a DNA sequence alignment between NMR19-16u and NMR19-4m elements with primer sequence information. Chop-PCR, small RNA blot, ChIP-PCR data rely on these sequences. They could show this alignment with representative accession form example Col, C24, Pu2-23

Response: We considered the reviewer’s suggestion and have aligned the DNA sequence of NMR19-16u and NMR19-4m elements with their primer sequences, and provided this information in supplementary Figure 5 (as shown below). In addition, we provided a new supplementary Table 5 that includes NMR19 sequences from all accessions which were used in the manuscript.

Supplementary Figure 5b. Alignment of NMR19 sequences. The black bar indicates enzyme cutting position for Chop-PCR; purple bar indicates the region used for ChIP-qPCR; red bar indicates the region used for small RNA Northern blot assay; green bar indicates the region used for copy number analysis.

(9) Supplementary Fig 4C: There is a typo: Tatol RNA -> Total RNA

Response: We apologize for this spelling mistake and thank the reviewer for pointing it out. We have corrected it in the revised supplementary Figure 4.

(10) Page 9: a typo Fig. 3b -> Fig. 4b

Response: We apologize for this spelling mistake and thank the reviewer for pointing

it out. We have corrected it in the revised manuscript (Line 243).

(11) 'Thus, gain or loss of methylation of NMR19-4 represents a 416 natural epimutation that is independent of genetic variation' Lin 416-417-> I am not sure if this statement is relevant or sufficient based on their datasets.

Response: We have changed this sentence to "Our genetic analysis failed to uncover any link between genetic variations and DNA methylation patterns in the examined NMRs, including NMR19." in the revised manuscript. (Lines 383-384).

(12) A typo in Supplementary Fig. 10a: label C24xrdm1 ddm1-15/F2 -> C24 x ddm1-15/F2?

Response: We apologize for this spelling mistake and thank the reviewer for pointing it out. We have corrected it in the revised version (Supplementary Figure 13).

Reviewers' comments:

Please note that while Reviewer 1 doesn't have Remarks to the Author, in his/her Remarks to the Editor, he/she says he/she is happy with the authors' responses to his/her comments and doesn't have further suggestions.

Reviewer #2 (Remarks to the Author):

The authors revised the manuscript quite well in response to reviewer's comments. In general however, the association between methylation and adaptation is still arguable at best, and it is important that the conclusions, and especially the abstract, should not be interpreted to mean that this study demonstrates epigenetic adaptation to climate change. From the discussion:

"We found that DNA methylation of NMR19-4 .. was associated with the 7 climate parameters in the Present time and 5 climate parameters in the Last Interglacial time (Supplementary Table 3), but PPH expression and leaf senescence did not show significant association with any climate parameters. Thus, DNA methylation at NMR19 was indeed involved in environmental adaptation of Arabidopsis, but it may not be the sole factor that is involved in the potential regulation of PPH expression by climate changes."

But actually, the fact that PPH expression is not associated, but methylation is associated with climate change could also mean that epigenetic factors have no impact on gene expression over time compared with genetic factors, and therefore have no adaptive consequences. This alternative needs to be spelled out in the discussion and the abstract.

Importantly only 6 unmethylated accessions are compared with six methylated accessions in the crucial Fig. 5b/c, and the phenotypes are weak, though significant. In Fig. 6a-c, there is no indication that the 2 accessions used in the F2 analysis have the same genetic haplotype at PPH, to exclude the possibility that linkage between epihaplotype and genetic haplotype could account for the correlation. Although the ddm1/C24 results suggest there is a difference between epialleles (6d-f), the effects are far less dramatic (6d vs 6b), indicating that there are other factors at work in the accessions.

Here are some specific comments.

It is not clear why the P values in 6f are lower than those in 6c, when the values and error bars look the other way round.

Line 111-112: "but the chromatin remodeler DDM1 may change a methylated NMR19-4 epiallele to an unmethylated NMR19-4 epiallele." I think the authors meant ddm1 mutation may change a methylated epiallele to an unmethylated one.

In Supplementary Figure 12, Chop-PCR is not quantitative as the authors confirmed that Chop-qPCR was able to distinguish the heterozygous epialleles of NMR19-4 in F1 ddm1-15 x C24 hybrid. DNA methylation of NMT19-4 may be also gradually lost in ddm1 background after generations.

Point-by-point response to reviewer comments

Reviewer #2 (Remarks to the Author):

The authors revised the manuscript quite well in response to reviewer's comments. In general however, the association between methylation and adaptation is still arguable at best, and it is important that the conclusions, and especially the abstract, should not be interpreted to mean that this study demonstrates epigenetic adaptation to climate change. From the discussion:

"We found that DNA methylation of NMR19-4 .. was associated with the 7 climate parameters in the Present time and 5 climate parameters in the Last Interglacial time (Supplementary Table 3), but PPH expression and leaf senescence did not show significant association with any climate parameters. Thus, DNA methylation at NMR19 was indeed involved in environmental adaptation of Arabidopsis, but it may not be the sole factor that is involved in the potential regulation of PPH expression by climate changes."

But actually, the fact that PPH expression is not associated, but methylation is associated with climate change could also mean that epigenetic factors have no impact on gene expression over time compared with genetic factors , and therefore have no adaptive consequences. This alternative needs to be spelled out in the discussion and the abstract.

Response: We agree with the Reviewer, and have spelled out the alternative possibility in the discussion and also reworded or toned down relevant statements in the abstract accordingly. The Discussion now reads " It is likely that DNA methylation of NMR19-4 is not the sole factor regulating *PPH* expression and leaf senescence in *A. thaliana* accessions, since complex regulatory mechanisms are involved in leaf senescence⁴⁰. On the other hand, NMR19-4 methylation might affect not only *PPH* expression but also the expression of other gene(s) that contribute to climate adaptation. Alternatively, the correlation between DNA methylation of NMR19-4 and climate data may not reflect an adaptive role for DNA methylation." (Lines 416-422). In the Abstract, we reworded some sentences: "In addition, further analysis indicated that DNA methylation of NMR19-4 correlates with local climates, implying that NMR19-4 is an environmentally associated epiallele. In summary, we discovered a novel epiallele, and provided some insights into its origin and potential function in local climate adaptation." (Lines 41-45).

Importantly only 6 unmethylated accessions are compared with six methylated accessions in the crucial Fig. 5b/c, and the phenotypes are weak, though significant. In Fig. 6a-c, there is no indication that the 2 accessions used in the F2 analysis have the same genetic haplotype at PPH, to exclude the possibility that linkage between epihaplotype and genetic haplotype could account for the correlation. Although the

ddm1/C24 results suggest there is a difference between epialleles (6d-f), the effects are far less dramatic (6d vs 6b), indicating that there are other factors at work in the accessions.

Response: In addition to the 6 unmethylated accessions, 6 methylated accessions and 5 deleted accessions shown in Fig. 5b/c, the leaf senescence phenotypes of the remaining 124 accessions were also determined (Supplemental Table 2). The entire leaf senescence dataset for all 141 accessions are displayed in Fig. 7a, which shows that there is a negative correlation between NMR19-4 methylation level and *PPH* expression and leaf senescence. We agree with the Reviewer that the phenotypic differences are not dramatic, so we cannot exclude that there may be other factors at work in the accessions. This point is now indicated in the results section: "Taken together, these findings suggest that NMR19-4m inhibits leaf senescence by repressing the expression of *PPH*, though genetic differences likely also influence the differences in leaf senescence and *PPH* expression between NMR19-4u accessions and NMR19-4m accessions." (Lines 301-305).

Here are some specific comments.

It is not clear why the P values in 6f are lower than those in 6c, when the values and error bars look the other way round.

Response: We apologize to the reviewer for this confusion. We carefully checked the original data and found that data of technical replicates was not considered while calculating P values in 6c. We have revised this to make it consistent.

Line 111-112: "but the chromatin remodeler DDM1 may change a methylated NMR19-4 epiallele to an unmethylated NMR19-4 epiallele." I think the authors meant ddm1 mutation may change a methylated epiallele to an unmethylated one.

Response: We thank the reviewer for catching this error. We have corrected this sentence to "but the mutation in a chromatin remodeler DDM1 may change methylated NMR19-4 epiallele to an unmethylated one." (Lines 110-112).

In Supplementary Figure 12, Chop-PCR is not quantitative as the authors confirmed that Chop-qPCR was able to distinguish the heterozygous epialleles of NMR19-4 in F1 ddm1-15 x C24 hybrid. DNA methylation of NMT19-4 may be also gradually lost in ddm1 background after generations.

Response: We agree with the reviewer. Therefore, in the revised manuscript we have included this possibility of gradual methylation loss in *ddm1* background after generations (Lines 245-246).

Reviewers' comments:

Reviewer #2 (Remarks to the Author):

The authors revised the manuscript satisfactorily in the abstract and discussion. Here are two suggestions to clarify the role of NMR-4 DNA methylation on PPH expression.

In Figure 7b, there is not a strong negative correlation between NMR-4 DNA methylation level and PPH expression. Adding statistical values such as p values in Figure 7a and 7b would be helpful to understand the relationship between these factors.

The second suggestion is to examine PPH RNA expression in F2 C24 x ddm1-15 population in Figure 4b. The authors could compare PPH expression of the wildtype plants with or without DNA methylation on NMR-4 locus. Since ddm1-15 has the same C24 background, this would be a good comparison to rule out genetic effects on PPH expression.

Point-by-point response to reviewer comments

Reviewer #2 (Remarks to the Author):

The authors revised the manuscript satisfactorily in the abstract and discussion. Here are two suggestions to clarify the role of NMR-4 DNA methylation on PPH expression.

In Figure 7b, there is not a strong negative correlation between NMR-4 DNA methylation level and PPH expression. Adding statistical values such as p values in Figure 7a and 7b would be helpful to understand the relationship between these factors.

Response: We thank the reviewer for this suggestion, and have added the corresponding p values in revised Figures 7a and 7b.

The second suggestion is to examine PPH RNA expression in F2 C24 x ddm1-15 population in Figure 4b. The authors could compare PPH expression of the wildtype plants with or without DNA methylation on NMR-4 locus. Since ddm1-15 has the same C24 background, this would be a good comparison to rule out genetic effects on PPH expression.

Response: We appreciate the reviewer's suggestion. In fact, we already performed this experiment in the F3 population (which had the same methylation status as in their respective F2 lines) of C24 x ddm1-15 and examined *PPH* expression and leaf senescence phenotype in these plants, and the data was presented in Figures 6d-6f and described in lines 293-301 of the Results section. As the reviewer indicated, since these different lines have the same genetic background, the results in Figures 6d-6f further confirmed that the methylation of NMR19-4 inhibits the expression of *PPH* and leaf senescence.

REVIEWERS' COMMENTS:

Reviewer #2 (Remarks to the Author):

From the previous rounds of reviews, the authors addressed whether DNA methylation status of NMR19-4 is associated with PPH expression and leaf senescence based on experimental evidence.

They observed the existence of epialleles at this locus in different Arabidopsis accessions. There was statistically significant but a weak negative correlation between DNA methylation of NMR-19 and PPH expression in Arabidopsis natural populations. However, there is still no direct evidence how much this PPH expression and leaf senescence contributed to the local climate adaption, as the authors stated few times, for example, in Line 353-357 "We found that DNA methylation of NMR19-4 was associated with 7 climate parameters in the Present time and 5 climate parameters in the Last Interglacial time (Supplementary Table 3), but PPH expression and leaf senescence did not show significant association with any climate parameters."

The other possible explanation would be that the climate conditions affected DNA methylation status of NMR19-4 and PPH expression but did not significantly contributed to local climate adaptation.

Therefore, we suggest the authors change the title accordingly by emphasizing the existence of NMR19 epiallele among Arabidopsis accessions and its effects on PPH expression and leaf senescence. For example "A Naturally Occurring Epiallele Controls Leaf Senescence and Contributes to Local Climate Adaptation in Arabidopsis" could be changed to "A Naturally Occurring Epiallele is associated with Leaf Senescence and with Local Climate Adaptation in Arabidopsis accessions"

- Line 469-470: "DDM1 might induce epialleles by affecting the inheritance of methylation in individual stochastic events." This part could mislead the readers. DDM1 protein is not an epimutation generator. Rather, the failure of DNA methylation maintenance in the DDM1-dependent pathway may cause epialleles.

Point-by-point response to reviewer comments

Reviewer #2 (Remarks to the Author):

From the previous rounds of reviews, the authors addressed whether DNA methylation status of NMR19-4 is associated with PPH expression and leaf senescence based on experimental evidence.

They observed the existence of epialleles at this locus in different Arabidopsis accessions. There was statistically significant but a weak negative correlation between DNA methylation of NMR-19 and PPH expression in Arabidopsis natural populations. However, there is still no direct evidence how much this PPH expression and leaf senescence contributed to the local climate adaption, as the authors stated few times, for example, in Line 353-357 “We found that DNA methylation of NMR19-4 was associated with 7 climate parameters in the Present time and 5 climate parameters in the Last Interglacial time (Supplementary Table 3), but PPH expression and leaf senescence did not show significant association with any climate parameters.”

The other possible explanation would be that the climate conditions affected DNA methylation status of NMR19-4 and PPH expression but did not significantly contributed to local climate adaptation.

Therefore, we suggest the authors change the title accordingly by emphasizing the existence of NMR19 epiallele among Arabidopsis accessions and its effects on PPH expression and leaf senescence. For example ““A Naturally Occurring Epiallele Controls Leaf Senescence and Contributes to Local Climate Adaptation in Arabidopsis” could be changed to “A Naturally Occurring Epiallele is associated with Leaf Senescence and with Local Climate Adaptation in Arabidopsis accessions”
Response: We changed the title as per reviewer's suggestion to "A Naturally Occurring Epiallele associates with Leaf Senescence and Local Climate Adaptation in Arabidopsis accessions".

Line 469-470: “DDM1 might induce epialleles by affecting the inheritance of methylation in individual stochastic events.” This part could mislead the readers. DDM1 protein is not an epimutation generator. Rather, the failure of DNA methylation maintenance in the DDM1-dependent pathway may cause epialleles.

Response: We agree with the reviewer's suggestion. Therefore, we have revised the statements to "The failure of DNA methylation maintenance in *ddm1* may cause epialleles" (Lines 471-472).